# The Role of Monk Parakeets as Nest-Site Facilitators in Their Native and Invaded Areas

**DOI:** 10.3390/biology10070683

**Published:** 2021-07-19

**Authors:** Dailos Hernández-Brito, Martina Carrete, Guillermo Blanco, Pedro Romero-Vidal, Juan Carlos Senar, Emiliano Mori, Thomas H. White, Álvaro Luna, José L. Tella

**Affiliations:** 1Department of Conservation Biology, Doñana Biological Station (CSIC), Calle Américo Vespucio 26, 41092 Sevilla, Spain; alvaro.luna@universidadeuropea.es (Á.L.); tella@ebd.csic.es (J.L.T.); 2Department of Physical, Chemical and Natural Systems, Universidad Pablo de Olavide, Carretera de Utrera, km 1, 41013 Sevilla, Spain; mcarrete@upo.es (M.C.); pedroromerovidal123@gmail.com (P.R.-V.); 3Department of Evolutionary Ecology, Museo Nacional de Ciencias Naturales (CSIC), José Gutiérrez Abascal 2, 28006 Madrid, Spain; g.blanco@csic.es; 4Museu de Ciències Naturals de Barcelona, Castell dels Tres Dragons, Parc Ciutadella, 08003 Barcelona, Spain; jcsenar@bcn.cat; 5Consiglio Nazionale delle Ricerche, Istituto di Ricerca sugli Ecosistemi Terrestri, Via Madonna del Piano 10, Sesto Fiorentino, 50019 Florence, Italy; moriemiliano@tiscali.it; 6Puerto Rican Parrot Recovery Program, US Fish and Wildlife Service, P.O. Box 1600 , Rio Grande, PR 00745, USA; thomas_white@fws.gov; 7Department of Health Sciences, Faculty of Biomedical and Health Sciences, Universidad Europea de Madrid, 28670 Madrid, Spain

**Keywords:** nest inquilines, biological invasions, cavity nesters, monk parakeet, facilitation, protective-nesting association, ecosystem engineer

## Abstract

**Simple Summary:**

Invasive species can be harmful to native species, although this fact could be more complex when some natives eventually benefit from invaders. Faced with this paradox, we show how the invasive monk parakeet, the only parrot species that builds its nests with sticks, can host other species as tenants, increasing nest-site availability for native but also exotic species. This same pattern is observed in the native range of the species, and when parakeets occupy urban or rural habitats, although the richness of tenants was higher in invaded areas and rural habitats. Tenants participated in the cooperative defense against predators, benefiting parakeets with their presence. As tenants can be both native and invasive species, management plans should consider the complex network of interactions developed with the invader.

**Abstract:**

While most of the knowledge on invasive species focuses on their impacts, little is known about their potential positive effects on other species. Invasive ecosystem engineers can disrupt recipient environments; however, they may also facilitate access to novel resources for native species. The monk parakeet (*Myiopsitta monachus*) is a worldwide invader and the only parrot that builds its own communal nests, which can be used by other species. However, the ecological effects of these interspecific interactions are barely known. We compared the role of the monk parakeet as a nest-site facilitator in different rural and urban areas, both invaded and native, across three continents and eight breeding seasons. A total of 2690 nests from 42 tenant species, mostly cavity-nesting birds, were recorded in 26% of 2595 monk parakeet nests. Rural and invaded areas showed the highest abundance and richness of tenant species. Multispecies communal nests triggered interspecific aggression between the monk parakeet host and its tenants, but also a cooperative defense against predators. Despite the positive effects for native species, monk parakeets also facilitate nesting opportunities to other non-native species and may also transmit diseases to tenants, highlighting the complexity of biotic interactions in biological invasions.

## 1. Introduction

Biotic interactions are important ecological and evolutionary drivers of community composition and richness [1]. The disruptive establishment of non-native species in new habitats may help us better understand how these ecological relationships among species develop and their consequences [2]. However, most studies on biotic relationships among non-native and native species have long been focused on antagonistic interactions (predation and competition), which are among the main factors of biodiversity loss [3,4,5,6]. Mutualistic and commensal interactions, which may be beneficial for both non-native and native species, have been poorly studied [7,8,9] and mostly focused on plants and invertebrates [7,10].

The monk parakeet (*Myiopsitta monachus*) is a successful and globally widespread avian invader, which has spread due to the international trade of thousands of wild-caught individuals from their native South American range [11,12,13]. Accidental escapes or releases of birds kept in captivity have founded several invasive populations [12,13], mainly in North America and Western Europe, but also in Asia, Africa, and some oceanic islands [12,13]. Unlike the other parrot species, monk parakeets build their own nests, a large structure of sticks located in trees or human construction such as pylons [14]. These nests normally include several chambers, each occupied by a different pair or kin groups that cooperate in the maintenance, breeding, and defense of the whole colony [14,15,16]. Monk parakeets usually tolerate the presence of other species in their nests, despite the fact that certain tenants may aggressively usurp active chambers [15]. The ability of monk parakeets to build nests that may be used by other vertebrate species [17,18,19,20,21,22,23] increases breeding site availability in a system, thus this species can be considered as an ecosystem engineer [24]. Ecosystem engineers are among the most influential forms of biological invasions due to their potential to affect native species via alterations in the abiotic environment. Moreover, ecosystem engineers can modify habitats and provide novel resources that may be exploited by other non-native species, increasing their establishment success [25]. These complex facilitative interactions could be considered as inquilinism, in which some species (nest inquilines or tenants) use nests of other species and they can show parasitic, commensalistic, and mutualistic interactions with their hosts [26,27]. However, to our knowledge, this kind of inquilinism has not been considered in birds, although similar interactions from other nest builder birds have been recorded [28,29,30].

Here, we investigate nest-site facilitation by the monk parakeet in its native and invaded ranges. We analyze which characteristics promote the presence of tenant species in monk parakeet nests, as well as interspecific interactions among tenants and monk parakeets. Our results highlight the role of monk parakeets as a provider of nest sites for different species in native but also in invaded areas, which benefit from the cooperative nest defense by monk parakeets and the tenants. We discuss the implications in the conservation of native communities of cavity nesters, and in the management of invasive species.

## 2. Materials and Methods

### 2.1. Data Recording and Field Procedures

From 2013 to 2020, we monitored populations of monk parakeets established in different countries including the native (Argentina, Brazil, Paraguay, and Uruguay) and invaded range (Italy, Puerto Rico, and Spain) of the species (Figure 1a) in urban and rural ranges. This monitoring was conducted in different campaigns of extensive field work that covered both rural and urban habitats in the native, e.g., [31] and invaded ranges [9,32,33,34] of monk parakeets, finding that the species is more abundant in rural areas in its native range, but proliferates more in urban areas in its invaded ranges. Once monk parakeet nests were located, we identified the substrate on which they were placed (tree or human construction), estimated their height above ground level (visually or using a laser range-finder for heights > 10 m), and counted the number of chambers as a measure of nest size. Regarding the type of substrate, we classified human constructions in two categories: pylon, which includes similar structures (e.g., poles and antenna towers), and roof, when nests are built on top of buildings. We observed the entrance of the different chambers (Figure 1b) using binoculars, recording evidence of breeding activity and the presence of monk parakeets or any other nesting species, hereafter called tenants. In some invasive populations, we were able to monitor the occupation of monk parakeet chambers by tenants during different breeding seasons (Madrid: 2014–2015, Seville: 2013–2019, Tenerife: 2014–2019). In these areas and Puerto Rico, we also assessed interspecific interactions by randomly sampling the behavior of different breeding monk parakeets for 15 min, following the same procedure used by Hernández-Brito et al. [32]. We recorded all species present within a radius of 15 m around the focal monk parakeet, and the existence (or not) of any aggressive interaction, which species started the attack, and which was the “winner” (i.e., that displaced the other). Similarly, we recorded interactions between monk parakeets and their tenants with predators. In all cases, we recorded the number of monk parakeets and tenants present and the habitat type (urban/rural).

### 2.2. Statistical Analysis

We compared the proportion of tenant nests belonging to native and non-native species in all recorded monk parakeet nests in both their native and invasive populations, using χ^2^ tests with Yate’s correction. We used Generalized Linear Models (GLM; [35]) to disentangle the factors affecting the probability of occupation of monk parakeet nests by tenant species (binomial error distribution, logit link function), and their richness and abundance (Poisson error distribution, log link function, corrected for zero-inflation) in the native and the invasive ranges of monk parakeets. In the case of populations monitored in multiple years, we selected the year with the highest number of monitored monk parakeet nests to avoid pseudoreplication, which resulted in a subset of all monk parakeet nests (50.36% of the total records). We also modeled factors influencing the persistence of tenants at monk parakeet chambers across years (number of occupied years over the total number of monitored years; binomial error distribution, logit link function), using only the populations monitored in consecutive years. As explanatory variables, we included the population (Continental Spain, Tenerife, Puerto Rico, and South America), habitat (urban/rural) and substrate (pylon, roof, or tree) where the colony was located, its size (i.e., number of chambers) and height above the ground, and the occupation of each chamber by at least one monk parakeet pair. For persistence models, we also included the number of monk parakeets breeding in each nest in the populations of Madrid, Seville, and Tenerife. Finally, we used GLM to assess the factors involved in the probability of aggressive interactions between monk parakeets and their tenants, and monk parakeets and their predators. For the first set of models, we fitted as explanatory variables the average body mass (in g) of the species interacting with monk parakeets (obtained from [36]), the number of monk parakeets and individuals of the interacting species involved, the habitat, and the population (Puerto Rico, Madrid, Seville, and Tenerife). When modeling interactions with predators, we also considered if monk parakeets were cooperating with individuals of other species during nest defense.

The model selection was performed using the Akaike Information Criterion corrected for small sample sizes, AICc [37]. Within each set of models (which includes the null model), we calculated the ΔAICc (as the difference between the AICc of model *i* and that of the best model) and the Akaike weight (*w*) of each model. Models within 2 AICc units of the best one were considered as alternatives and used to perform model averaging (package MuMIn; [38]). We considered that a given effect received no, weak, or strong support when the 95% confidence interval (CI) strongly overlapped zero, barely overlapped zero, or did not overlap zero, respectively. Post-hoc tests were performed using the *emmeans* package [39], applying the Bonferroni correction. All statistical analyses were conducted in R v. 4.0.3 [40].

## 3. Results

### 3.1. Monk Parakeet Nests and Their Communities of Tenants

We monitored 2595 monk parakeet nests, 1869 in the invaded and 726 in the native range of the species. Most nests in the invaded area were located in urban habitats (96.36%), while native ones were mainly in rural habitats (84.02%). We recorded 2690 nests of 42 tenant species (40 birds, one mammal and one social insect; Table 1) nesting in 26.35% of the monk parakeet nests. These species, of which 47.62% were secondary cavity nesters which accumulated most (82.55%) of the recorded tenant nests (Table 1), showed three different nesting strategies. First, the external stick conglomerate of monk parakeet nests was used as a nesting substrate in which tenants added extra vegetal nest material or built a loosely woven nest of vegetal fibers on the structure (Figure 1c). Second, tenants used small openings in the stick conglomerate that were independent of nesting chambers built by monk parakeets (Figure 1d). Third, tenants occupied the nesting chambers built by monk parakeets (Figure 1e). Multiple strategies could be present at the same time in a single colonial structure, favoring species coexistence. Indeed, we recorded up to six different tenant species cohabiting with monk parakeets in a single colony (i.e., Feral pigeon *Columba livia* var *domestica*, Western jackdaw *Coloeus monedula*, House sparrow *Passer domesticus*, Spanish sparrow *P. hispaniolensis*, Eurasian tree sparrow *P. montanus*, and Spotless starling *Sturnus unicolor*). The highest concentration of tenant nests in a single monk parakeet nest (Figure 1b) included 35 nests of House and Eurasian tree sparrow, spotless starling, and stock dove *Columba oenas*.

The proportion of tenant nests of non-native species was much lower in the areas invaded by monk parakeets (2.95%, *n* = 2436 tenant nests) than in their native areas (28.86%, *n* = 194 tenant nests) (χ^2^ = 254.98, *p* < 0.0001). The presence, abundance, and richness of tenants at monk parakeet nests were higher in all invasive populations compared to the native range (Figure 2; Table 2; Appendix A; Appendix A). Monk parakeet nests most likely to host tenants and with a more abundant and richer community of tenant species were those with more chambers (i.e., larger nests), located on roofs and rural habitats. Further, abandoned nests held more tenants (in terms of the number of nesting pairs and species) than occupied ones. The annual monitoring of monk parakeet nests and their tenants in Seville, Tenerife, and Madrid showed that, once settled, tenants persisted more years in the monk parakeet nest when it was located in large, rural nests occupied by a low number of monk parakeets and placed at a low height (Table 3 and Appendix A).

### 3.2. Interactions between Monk Parakeets and Tenant Species

We recorded 535 encounters between monk parakeets and individuals of 19 tenant species that approached within ≤15 m of the breeding colony (Figure 3). Only 21.53% of these encounters resulted in aggressions and were mainly initiated by monk parakeets (65.57%). Less than half of these aggressions (48.70%) resulted in the expulsion of tenants from the colonies. Neither the number of individuals nor the body size of the tenants affected the probability of developing or winning an aggressive encounter with monk parakeets (Table 4).

We documented 21 nest usurpations by four different tenant species, namely: the Rose-ringed parakeet (*Psittacula krameri*), the Eurasian kestrel (*Falco tinnunculus*) (see picture in Figure 3), the American kestrel (*Falco sparverius*), and the stock dove. These usurpations typically involved short events of aggressions during 1–7 days, until monk parakeets finally lost their chamber. In only one case, we detected dead monk parakeet nestlings after the occupation of a contiguous chamber by Eurasian kestrels. Moreover, we observed 31 events of cooperative defense of colonies between tenant species and monk parakeets against potential predators such as common buzzards (*Buteo buteo*) (16.13%), red-tailed hawks (*Buteo jamaicensis*) (12.9%), common ravens (*Corvus corax*) (12.9%), feral cats (*Felis catus*) (3.23%), Mediterranean yellow-legged gulls (*Larus michahellis*) (9.68%), black kites (*Milvus migrans*) (12.9%), and black rats (*Rattus rattus*) (32.26%). Indeed, cooperation between monk parakeets and their tenants was the only factor explaining the probability of successfully expelling a predator from the proximity of a nest (Table 4).

## 4. Discussion

Understanding biotic interactions between invasive species and their recipient communities is crucial to determine the success of biological invasions and predict their ecological effects [2]. Most studies have been focused on interactions with a detrimental output for one of the interacting species, such as predation or competition, which often lead to native population declines, reduced biodiversity, and altered ecosystem functioning [8]. Facilitative processes, however, have been less explored and focused mainly on animal–plant interactions, such as seed dispersal or pollination [41]. However, positive interactions are expected to be widespread across taxa [42]. Here, we show how monk parakeets can influence these interactions through nest-site facilitation for other species. This facilitation, which is not only present in the native range of the species but also in its invasive populations, results in a “nest web” where monk parakeets act as primary cavity nesters that provide a limiting resource to several secondary cavity nesters [43], thus assuming the role of an ecosystem engineer. The high nest-site tenacity recorded for tenants, which was higher for pairs using large nests with a high number of empty chambers, confirms that monk parakeet nests are a valuable resource for many species that use them recurrently as tenants. Moreover, other non-cavity nester species can also associate with these breeding cores (see also [19,21,23]), using monk parakeet nests as substrate.

Besides nesting opportunities, there are other benefits derived from this association between monk parakeets and their tenants. Monk parakeet nests can offer microclimate control (thermoregulation) that may be key in determining avian reproductive success [8,44,45], as temperatures inside the nest can affect the survival and growth of nestlings [46]. These microclimate conditions in monk parakeet nests, combined with the colonial life and the reuse of chambers, can also promote a high parasite load that may negatively affect the breeding success of birds [47]. Moreover, in the invaded range, monk parakeets can acquire parasites from the recipient community and, in turn, can potentially introduce novel parasites and diseases into the recipient community [33,48,49,50,51,52,53]. However, monk parakeets seem to be more resistant to infection by local parasites than native species [33], thus this transference of parasites could be strongly weighted towards native species. Additionally, Viana et al. [54] have found that, in the native range, the use of certain plants such as *Eucalyptus* spp. for nest building has bactericidal effects that can inhibit the growth of pathogenic bacteria in the nest, likewise another parrot species [55]. Although there are no studies about this topic in the invaded range, the widespread distribution of introduced eucalypt species and their use for nest building in other areas suggest that this effect can potentially be extended across the entire distribution of monk parakeets.

Despite the microclimatic benefits associated with communal nesting, these aggregations of species may trigger a high degree of interspecific competition [56] that can increase breeding failure [21]. Although we detected aggressive encounters among monk parakeets and tenants, their occurrence was relatively low and mainly initiated by monk parakeets in an attempt to avoid the usurpation of active chambers. However, interspecific nesting associations at monk parakeet nests may be important to reduce predation risk through cooperative defense [57,58,59]. Considering that tenants were more frequent and abundant in rural than in urban areas where predators are more common and abundant, these interspecific associations may help monk parakeets spread from urban into more natural areas. Similar interspecific protective associations between monk parakeets and a native bird species (White stork, *Ciconia ciconia*) have been previously described as important for the expansion of this invasive species into rural habitats [9]. Consequently, the reduction of nest predation may increase the fledging success of monk parakeets [34] as well as the economic impacts of this species on the agriculture in its invaded rural areas [60].

Tenants using monk parakeet nests include non-native species, which were more frequent in the native than in the invaded range of the species. Of special concern is the presence of rose-ringed parakeets at monk parakeet nests. The rose-ringed parakeet is one of the most successful avian invaders, and which needs cavities for nesting. By providing this limiting resource, monk parakeets may also be favoring the establishment and spread of rose-ringed parakeets, constituting an example of facilitative non-native species interactions (e.g., invasional meltdown hypothesis; [8,23,61]). However, monk parakeets may not only assist the establishment of invasive but also threatened [62] or rare native species. For example, the recent expansion of western jackdaws and stock doves in urbanized areas of Madrid coincides with monk parakeet population growth and expansion (G.B. pers. obs.). These native species are common tenants of monk parakeet nests (Table 1), and this facilitation could be positively affecting their declining populations [63,64,65]. Thus, synergic interactions between native and invasive species may have effects not only on the invasive success of monk parakeets, but also native and invasive tenant species [66]. These side-effects associated with invasive species have been undervalued despite their implications for invasion ecology and native vertebrate communities [67]. Therefore, these should be considered during management plans focused on population control measures (i.e., removal of nests) of monk parakeets, especially in Mediterranean regions [68], to avoid unexpected results or negative consequences for native species [69,70].

## 5. Conclusions

A poor understanding of interspecific interactions in biological invasions reduces our ability to predict and detect impacts on the recipient communities [70]. Here, using an unprecedented dataset collected in several areas within the native and invaded range of monk parakeets, we show that these parrots can be considered as ecosystem engineers that provide nest substrates to many other species that overlap their spatial distribution with them. Biotic interactions between monk parakeets and their tenants constitute a complex web that includes positive relationships, but which can also trigger negative consequences for tenant communities, especially in areas where monk parakeets are invaders. Thus, the urgent implementation of control actions on invasive monk parakeet populations should consider this balance of cost–benefits to minimize impacts on biodiversity. We hope that our study contributes towards the knowledge of invasion science in the current scenario of global change.

## Figures and Tables

**Figure 1 biology-10-00683-f001:**
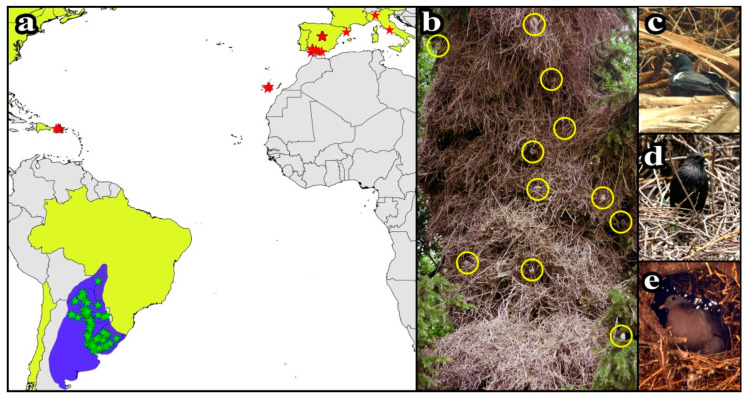
(**a**) Location of monk parakeet nests monitored in the native (blue area; green stars) and invaded (yellow area; red stars) ranges of the species. (**b**) Monk parakeet colony established in a rural area (Madrid, Spain) with tenant nests (yellow circles) of two sparrow species (house sparrow *Passer domesticus* and Eurasian tree sparrow *P. montanus*). Different nesting strategies by tenant species: (**c**) greater Antillean grackle *Quiscalus niger* using monk parakeet nests as nesting substrate, (**d**) spotless starling *Sturnus unicolor* nesting in openings from external stick conglomerate, and (**e**) stock dove *Columba oenas* occupying chambers built by monk parakeets. Photos: Dailos Hernández-Brito.

**Figure 2 biology-10-00683-f002:**
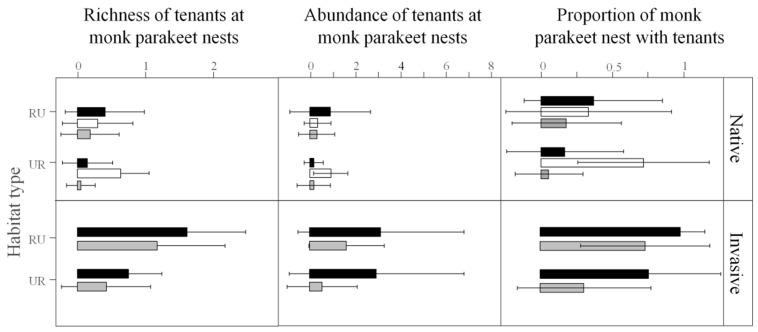
Proportion of nests (±SD) with tenants, and abundance (±SD) and richness (±SD) of tenants in the native and invasive populations of monk parakeets *Myiopsitta monachus*. Bars represent nests located in pylons (black bars), roofs (white bars), and trees (grey bars), grouped by urban (UR) and rural (RU) habitats.

**Figure 3 biology-10-00683-f003:**
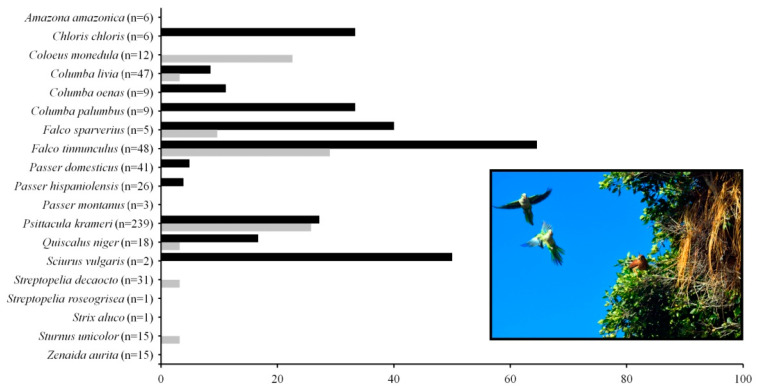
Percentage of encounters between monk parakeets and different tenant species that ended in aggressions (black bars) and percentage of cooperative defense between tenant species and monk parakeets against predators (grey bars). The number of recorded encounters between tenants and monk parakeets is shown in brackets. The inset picture shows a pair of Eurasian kestrels *Falco tinnunculus* usurping a chamber despite the active defense of monk parakeets. Photo: Dailos Hernández-Brito.

**Table 1 biology-10-00683-t001:** List of recorded tenant species and their number of nesting pairs present in monk parakeet nests in the native (South America: Argentina, Brazil, Paraguay, Uruguay) and invaded (Italy, Puerto Rico, Continental Spain, Tenerife) ranges of monk parakeets. The number of recorded tenant nests used in GLMs (shown in Table 2) are in parentheses.

Tenant Species	Invaded Areas				Native Areas				Total Records
	Italy	P. Rico	C. Spain	Tenerife	Argentina	Brazil	Paraguay	Uruguay	
**BIRDS**									
*Agelaioides badius* ^V,O/C^					4 (4)			1 (1)	5
*Amazona amazonica* ^V,C^				1 *					1
*Anumbius annumbi* ^S^					2 (2)				2
*Caracara plancus* ^P,S^					6 (2)				6
*Chloris chloris* ^S^			11 (6)						11
*Ciconia ciconia* ^P,S^	2								2
*Columba livia* var. *domestica* ^V,C^		4 (4) *	185 (86)	4	7 (7) *				200
*Columba oenas* ^V,C^			268 (142)						268
*Columba palumbus* ^S^			6 (2)						6
*Coloeus monedula* ^V,P,C^			196 (123)						196
*Coryphistera alaudina* ^C^							1 (1)		1
*Dendrocygna autumnalis* ^V,C^					1 (1)				1
*Falco sparverius* ^V,P,C^		2 (2)			3 (2)	1 (1)		1 (1)	7
*Falco tinnunculus* ^V,P,C^			20 (12)	9 (1)					29
*Geranoaetus polyosoma* ^P,S^					2				2
*Machetornis rixosa* ^O/C^					10 (10)	1 (1)	2 (2)	1 (1)	14
*Mimus saturninus* ^C^								1 (1)	1
*Molothrus bonariensis* ^C^		1 (1) * †							1
*Otus scops* ^V,P,C^			1						1
*Passer domesticus* ^V,O/C^		27 (27) *	852 (492)		94 (87) *		10 (10) *	33 (33) *	1016
*Passer hispaniolensis* ^S^			285 (133)	39 (4)					324
*Passer italiae* ^V,O/C^	4								4
*Passer montanus* ^V,O/C^			278 (122)						278
*Pitangus sulphuratus* ^S/C^					3 (2)	2(1)		4 (4)	9
*Psittacara leucophthalmus* ^V/C^							1 (1)		1
*Patagioenas picazuro* ^S^					1				1
*Psittacula krameri* ^V/C^				35 (6) *					35
*Quiscalus niger* ^S^		42 (42)							42
*Schoeniophylax phryganophilus* ^S/C^					2 (2)				2
*Sicalis flaveola* ^V,O/C^					4 (4)	1 (1)		7 (7)	12
*Sicalis luteola* ^O/C^					2 (2)				2
*Streptopelia decaocto* ^S/C^			28 (10)	4					32
*Streptopelia roseogrisea* ^S^				4 *					4
*Strix aluco* ^V,P,C^			1 (1)						1
*Sturnus unicolor* ^V,O/C^			158 (100)						158
*Thraupis palmarum* ^C^							2 (2)		2
*Tyrannus melancholicus* ^C^							1 (1)		1
*Tyrannus savana* ^S^					1				1
*Upupa epops* ^V,C^			1						1
*Zenaida aurita* ^S^		3 (3)							3
**MAMMALS**									
*Sciurus vulgaris* ^V,P,C^			4 (2)						4
**INSECTS**									
Unknown bee (Apoidea) ^V,C^						1 (1)			1
**UNKNOWN TENANT**					1 (1)		1 (1)		2

Traits of tenants: (^V^) a cavity nester, (^P^) a predator. Nest use: how tenant species used colonies: as nesting substrate (^S^), occupying openings (^O^), or occupying nesting chambers (^C^). * Tenant species is a non-native species in the area. † Brood parasite species that parasitized on a house sparrow’s egg-laying.

**Table 2 biology-10-00683-t002:** Models obtained to assess the effects of different variables (see footer) on the presence, abundance, and richness of tenant species at monk parakeet nests. Estimates and 95% confidence intervals (2.5% and 97.5%) were obtained after model averaging. k: number of parameters, ΔAICc: difference between the AICc (i.e., the Akaike Information Criterion corrected for small sample sizes) of model i and that of the best model (i.e., the model with the lowest AICc), w: Akaike weights. The fit of the model including all variables used in model averaging can be checked in Appendix A. In bold, variables receiving strong support (i.e., the 95% confidence interval did not overlap with zero). The first ten models obtained are shown.

**Models of Presence**	***k***	**ΔAICc**	***w***	**Variables**	**Estimate**	**2.50%**	**97.50%**
A + B + C + D + E	9	0.00	0.73	A	22.77	−6219.85	6265.39
A + B + C + F + D + E	10	1.95	0.27	**B**	**1.59**	**1.40**	**1.78**
A + B + D + E	8	16.91	0.00	**C**(**urban**)	**−1.39**	**−2.05**	**−0.74**
A + B + F + D + E	9	18.91	0.00	**D**(**Puerto Rico**)	**3.14**	**2.50**	**3.77**
A + B + C + D	7	68.19	0.00	**D**(**C. Spain**)	**2.94**	**2.23**	**3.65**
A + B + C + F + D	8	69.62	0.00	**D**(**Tenerife**)	**1.27**	**0.03**	**2.51**
A + B + D	6	69.94	0.00	**E**(**roof**)	**3.89**	**2.58**	**5.20**
A + B + F + D	7	71.51	0.00	E(tree)	−0.66	−1.34	0.03
A + B + C + E	6	123.91	0.00	F	0.02	−0.11	0.15
A + B + C + F + E	7	125.30	0.00				
**Models of Abundance**	***k***	**ΔAICc**	***w***	**Variables**	**Estimate**	**2.50%**	**97.50%**
A + B + C + D + E	11	0.00	0.72	**A**	**1.54**	**1.28**	**1.80**
A + B + C + F + D + E	12	1.85	0.28	**B**	**0.87**	**0.79**	**0.95**
A + B + D + E	10	44.58	0.00	**C**(**urban**)	**−0.97**	**−1.25**	**−0.69**
A + B + F + D + E	11	45.17	0.00	**D**(**Puerto Rico**)	**1.80**	**1.43**	**2.17**
A + B + C + D	9	57.59	0.00	**D**(**C. Spain**)	**1.67**	**1.37**	**1.98**
A + B + C + F + D	10	59.61	0.00	D(Tenerife)	0.61	−0.14	1.36
A + B + F + D	9	121.88	0.00	**E**(**roof**)	**1.40**	**0.70**	**2.10**
A + B + D	8	122.45	0.00	**E**(**tree**)	**−0.68**	**−0.99**	**−0.37**
B + C + F + D + E	11	136.62	0.00	F	0.02	−0.06	0.10
B + C + D + E	10	137.88	0.00				
**Models of richness**	***k***	**ΔAICc**	***w***	**Variables**	**Estimate**	**2.50%**	**97.50%**
A + B + C + D + E	10	0.00	0.73	**A**	**1.07**	**0.85**	**1.29**
A + B + C + F + D + E	11	2.01	0.27	**B**	**0.32**	**0.29**	**0.35**
A + B + C + D	8	30.82	0.00	**C**(**urban**)	**−0.85**	**−1.10**	**−0.61**
A + B + C + F + D	9	32.46	0.00	**D**(**Puerto Rico**)	**1.20**	**0.86**	**1.55**
A + B + D + E	9	44.19	0.00	**D**(**C. Spain**)	**1.43**	**1.14**	**1.72**
A + B + F + D + E	10	45.66	0.00	**D**(**Tenerife**)	**1.79**	**1.32**	**2.27**
A + B + D	7	55.59	0.00	**E**(**roof**)	**1.94**	**1.35**	**2.52**
A + B + F + D	8	57.57	0.00	E(tree)	0.19	−0.08	0.45
B + C + D + E	9	78.17	0.00				
B + C + F + D + E	10	79.45	0.00				

A: abandoned nest (i.e., without presence of monk parakeets), B: nest size (i.e., number of chambers), C: habitat type (urban/rural), D: population (South America, Puerto Rico, Continental Spain, and Tenerife), E: substrate (tree, pylon, roof), F: height above ground level.

**Table 3 biology-10-00683-t003:** Models obtained to assess the effects of different variables (see footer) on the persistence (years that a nest remained occupied) of tenants at monk parakeet nests. Estimates and 95% confidence intervals (2.5% and 97.5%) were obtained after model averaging. *k*: number of parameters, ΔAICc: difference between the AICc (i.e., the Akaike Information Criterion corrected for small sample sizes) of model i and that of the best model (i.e., the model with the lowest AICc), *w*: Akaike weights. The fit of the model including all variables used in model averaging can be checked in Appendix A. In bold, variables receiving strong support (i.e., the 95% confidence interval did not overlap with zero). The first ten models obtained are shown.

Model	*k*	ΔAICc	*w*	Variables	Estimate	2.50%	97.50%
G + A + B + C + F	6	0.00	0.51	**G**	**−7.98**	**−8.92**	**−7.03**
G + A + B + C + F + E	7	1.54	0.24	**A**	**2.03**	**0.93**	**3.13**
G + A + B + C + F + D	8	2.18	0.17	**B**	**9.88**	**8.84**	**10.92**
G + A + B + C + F + D + E	9	3.77	0.08	**C**(**urban**)	**−5.30**	**−7.95**	**−2.66**
G + B + C + F	5	16.70	0.00	**F**	**−0.48**	**−0.62**	**−0.33**
G + B + C + F + E	6	18.65	0.00	E(tree)	0.97	−1.84	3.78
G + B + C + F + D	7	18.99	0.00				
G + B + C + F + D + E	8	20.97	0.00				
G + A + B + C + D	7	30.39	0.00				
G + A + B + F + E	6	31.20	0.00				

A: Abandoned nest (i.e., without presence of monk parakeets), B: nest size (i.e., number of chambers), C: habitat type (urban/rural), D: population (Seville, Madrid, and Tenerife), E: substrate (tree, pylon, roof), F: height above ground level, G: number of monk parakeets in the colony.

**Table 4 biology-10-00683-t004:** Models obtained to assess factors involved in the probability of developing aggressive interactions or winning them when monk parakeets interacted with their tenants and in the probability of expelling predators through aggressive interactions. Estimates and 95% confidence intervals (2.5% and 97.5%) were obtained after model averaging. *k*: number of parameters, ΔAICc: difference between the AICc (i.e., the Akaike Information Criterion corrected for small sample sizes) of model i and that of the best model (i.e., the model with the lowest AICc), *w*: Akaike weights. The fit of the model including all variables used in model averaging can be checked in Appendix A. In bold, variables receiving strong support (i.e., the 95% confidence interval did not overlap with zero). The first ten models obtained are shown.

**Aggressive Interactions**	***k***	**ΔAICc**	***w***	**Variables**	**Estimate**	**2.50%**	**97.50%**
T	4	0.00	0.25	T(Puerto Rico)	−0.89	−2.03	0.25
U + T	5	0.09	0.24	T(Seville)	−1.14	−2.37	0.10
V + T	5	1.57	0.11	T(Tenerife)	0.63	−0.23	1.49
U + V + T	6	1.65	0.11	U	0.05	−0.02	0.13
W + T	5	1.72	0.11	V(urban)	0.41	−0.76	1.58
W + U + T	6	1.87	0.10	W	0.00	0.00	0.00
W + V + T	6	3.43	0.04				
W + U + V + T	7	3.56	0.04				
V	2	22.90	0.00				
U + V	3	24.33	0.00				
**Winning an Aggression**	***k***	**delta**	***w***	**Variables**	**Estimate**	**2.50%**	**97.50%**
W + T	5	0.00	0.16	W	0.01	0.00	0.01
U + T	5	0.53	0.12	T(Puerto Rico)	−0.02	−3.19	3.14
W + V	3	0.72	0.11	T(Seville)	−1.33	−4.32	1.66
W + U + T	6	0.83	0.11	T(Tenerife)	−2.04	−4.33	0.25
W	2	1.25	0.09	U	0.09	−0.04	0.22
T	4	1.25	0.08	V(urban)	−1.68	−3.97	0.60
W + U + V	4	1.76	0.07				
W + V + T	6	2.10	0.06				
W + U	3	2.35	0.05				
U + V + T	6	2.59	0.04				
**Expelling a Predator**	***k***	**delta**	***w***	**Variables**	**Estimate**	**2.50%**	**97.50%**
T + X	6	0.00	0.15	T(Madrid)	−19.87	−25,255.80	25,216.06
T + Y	6	1.53	0.07	T(Puerto Rico)	−18.21	−25,254.14	25,217.72
T + Y + X	7	1.78	0.06	T(Seville)	−18.78	−25,254.71	25,217.15
V + X	3	1.98	0.05	T(Tenerife)	−18.53	−25,254.46	25,217.40
T + V + X	7	1.99	0.05	**X**	**2.23**	**0.66**	**3.79**
X	2	2.09	0.05	Y	1.30	−0.77	3.37
T + Z + X	7	2.23	0.05	V(urban)	0.56	−0.66	1.78
T + U + X	7	2.31	0.05				
Y	2	3.19	0.03				
U + V + X	4	3.27	0.03				

T: population (Madrid, Puerto Rico, Seville, and Tenerife), U: number of interacting monk parakeets, V: habitat type (urban/rural), W: body mass of the interacting tenant species, X: cooperation (i.e., cooperative defense between monk parakeet hosts and tenant species), Y: number of interacting tenants, Z: body mass of the predator.

## Data Availability

The raw data supporting the findings of this article have been uploaded as Appendix A. The rest of the raw data is provided in the body of the article.

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
