# Peer review of "The Role of Monk Parakeets as Nest-Site Facilitators in Their Native and Invaded Areas"

_biology, 2021, doi:10.3390/biology10070683_

Round 1

Reviewer 1 Report

This manuscript is generally well written and was a pleasure to read.  The following comments are provided.

Care needs to be taken in deciding when to use the generic term 'parakeets' rather than 'monk parakeets'.  In nearly all cases I think the later term should be used.  Suggest doing a global find-change search and making the appropriate adjustment(s).

L27 Change 'from' to 'with'

L29 Change 'by' to 'with'

L32 Add 'access to' after '... also facilitate ...'

L39 Change 'tenants' to 'tenant species.'

L40 Change 'aggressions' to 'aggression'

L41 Change 'facilitated' to 'facilitate'

L42 Change 'nest sites' to 'nesting opportunities'

L50 Change 'emergence' to 'establishment' and 'into' to 'in'

L63 Change 'Contrary to' to 'Unlike'

L64 Change 'on' to 'in'

L77 Change 'areas' to 'range'

L81 Change 'of' to 'by' and 'with other ..." to 'and the'

L100 Change 'an' to 'any'

L102 Change 'annotate' to 'recorded'

L114 Add 'in' after 'monk parakeet nests'

L132 Change 'fit' to 'fitted'

L168 Add 'to' after 'up'

Figure 2.  Suggest you enlarge the whole figure, as it is difficult to see the detail in the cells.

L230-231 Falco tinnunuculus needs to be in italics

L237 Add 'did' after '... case,..' and change 'detected' to 'detect'

L285 I can't understand how Monk parakeets could introduce 'novel' parasites and diseases into their native community.  Certainly that could happen in the invaded range.

L287 Change 'natives' to 'native species'

L289 The observation of monk parakeets using eucalypt branches in nest construction, and the possible pathogen control benefits that flow from that , make this species only the second recorded to do so.  The first was the galah (Eolophus roseicapillus) - suggest reading Rowley, I. (1990) Galah. : Behavioural Ecology of Galahs. Surrey Beatty and Sons Pty Ltd. [pgs 77-79].

L290 Change Eucaliptus' to 'Eucalyptus'

L310 It is not clear which 'species' is being referred to here.

L318 Change 'increment of' to 'increase in'

 L27 Change 'implantation' to 'implementation'

L400 and L439 Change 'bull.' to 'Bull.'

Author Response

Reviewer 1:

This manuscript is generally well written and was a pleasure to read.  The following comments are provided.

Thank you very much for these positive comments, we appreciate very much all the detailed suggestions provided. We have addressed all of them since we feel they are helping us to improve the clarity of our manuscript. Changes can be easily seen in the new version with tracked changes.

Care needs to be taken in deciding when to use the generic term 'parakeets' rather than 'monk parakeets'.  In nearly all cases I think the later term should be used.  Suggest doing a global find-change search and making the appropriate adjustment(s).

We have changed the generic term "parakeets" to "monk parakeets" throughout the manuscript.

L27 Change 'from' to 'with'

Done.

L29 Change 'by' to 'with'

Done.

L32 Add 'access to' after '... also facilitate ...'

Done.

L39 Change 'tenants' to 'tenant species.'

Done.

L40 Change 'aggressions' to 'aggression'

Done.

L41 Change 'facilitated' to 'facilitate'

Done.

L42 Change 'nest sites' to 'nesting opportunities'

Done.

L50 Change 'emergence' to 'establishment' and 'into' to 'in'

Done.

L63 Change 'Contrary to' to 'Unlike'

Done.

L64 Change 'on' to 'in'

Done.

L77 Change 'areas' to 'range'

Done.

L81 Change 'of' to 'by' and 'with other ..." to 'and the'

Done.

L100 Change 'an' to 'any'

Done.

L102 Change 'annotate' to 'recorded'

Done.

L114 Add 'in' after 'monk parakeet nests'

Done.

L132 Change 'fit' to 'fitted'

Done.

L168 Add 'to' after 'up'

Done.

Figure 2.  Suggest you enlarge the whole figure, as it is difficult to see the detail in the cells.

We have redone this figure for clarity.

L230-231 Falco tinnunuculus needs to be in italics

Done.

L237 Add 'did' after '... case,..' and change 'detected' to 'detect'

Done.

L285 I can't understand how Monk parakeets could introduce 'novel' parasites and diseases into their native community.  Certainly that could happen in the invaded range.

We did not try to suggest that monk parakeets introduce novel parasites in their native range but invasive monk parakeets introduce them in their recipient communities. We have changed "native community" to "recipient community" to clarify this point (line 307).

L287 Change 'natives' to 'native species'

Done.

L289 The observation of monk parakeets using eucalypt branches in nest construction, and the possible pathogen control benefits that flow from that, make this species only the second recorded to do so.  The first was the galah (Eolophus roseicapillus) - suggest reading Rowley, I. (1990) Galah. : Behavioural Ecology of Galahs. Surrey Beatty and Sons Pty Ltd. [pgs 77-79].

Thank you for your suggestion, we have added this cite in the text (line 313)

L290 Change Eucaliptus' to 'Eucalyptus'

Sorry for the spelling mistake. Done.

L310 It is not clear which 'species' is being referred to here.

We have changed "This species" to "The rose-ringed parakeet" to clarify it.

L318 Change 'increment of' to 'increase in'

Done. As both reviewers provided different alternatives to change parts of this sentence, we have chosen the option from the 2nd reviewer.

L27 Change 'implantation' to 'implementation'

Sorry for the spelling mistake. Done.

L400 and L439 Change 'bull.' to 'Bull.'

Done.

Reviewer 2 Report

This is an impressive data set on a specific and interesting aspect of the biology of an invasive species. The authors present well-documented data that Monk Parakeets are ecosystem engineers creating nesting sites for other species by virtue of their unusual nest structure and behavior.  The paper is well written and the analyses well structured.  The figures are useful (the photos added will help readers unfamiliar with Monk Parakeet nests).  Below are few questions and suggestions that will make the paper clearer.  

While the authors state that Monk Parakeets are one of the most widespread and successful invasive bird species, their success in terms of going outside of specific urban areas in most (all?) of the regions where they have become established pales in comparison to other bird species like European Starling, House Sparrow and Eurasian Collared Dove (at least from a North American perspective) so this language could be tempered a little. 

With respect to how were nests searched for and identified, do you worry about any spatial biases in sampling locations (specifically, data composed mostly of urban nests in invasive range and mostly rural nests in native range)? 

Were nests in different regions of the world visited during comparable periods of time (other than the previously stated nests visited during the breeding season)? Do you think the species composition of tenants might be affected by seasonal changes in biodiversity and nest usage (due to migratory species)? 

Other suggestions:

Line 32.  Replace "facilitate" with "present"

Line 44. Keywords:  "nest inquilines" I looked up this term and can see why it is used, but it is never brought up anywhere else in the paper and I haven't seen it used with birds before.  I think it should be brought up and defined somewhere in the introduction.  

Also: are many of these "cavity nesters" or species that like to place nests in cavities (e.g., Passer)?  There may not be a better term here.

Line 64: “Human utilities”, should be "human-made structures"  or "human construction"

Line 87.  Change "areas" to "ranges"

Line 89: As Line 64, should be human structure or man-made structure .

Line 98.  I think the word "conservatively" can be deleted here.

Line 106.  Say where the photo was taken.

Line 121.  Change ", obtaining thus" to "which resulted in..." 

Lines 124-126: Contrary to Lines 93-95, it seems to be implied that nests in Puerto Rico were also sampled across multiple years, in order to be included in the persistence analysis. If so, what years were sampled?  

Line 127: What is meant by “roof” as a substrate? Were nests built on top of ledges, on windowsills, under bridges, (and other man-made structures that are not pylons) also labeled as roof? I would suggest rewording to use a clearer term for “roof” or an in-text clarification of what this category encompasses. 

Lines 131-132: …between monk parakeets and their tenants, and monk parakeets and their predators. 

Line 166.  This could be made a little clearer (and maybe tested).  Were there likely to be more species using a nest structure if species had different needs (nest types, one.

Line 266: Should the word cooperative be used instead of “facilitative”? 

Line 315. Change to "parakeets"

Line 318.  Change this to "coincides with monk parakeet population growth and expansion.

Line 320.  Any idea why these species are declining?

Line 324.  Make the context of this clear, are you meaning if management plans dictate population control measures?

Table 1.  Make the distinction between invaded and native areas clear in the heading.

Figure 2 is referenced in Line 191 to show that invasive populations had a higher presence, abundance, and richness of tenants than the native range. The graphs in the figure do not make that point obvious (seeing all the overlap among the confidence intervals).  This figure could be moved to the supplemental data.

Author Response

Reviewer 2:

This is an impressive data set on a specific and interesting aspect of the biology of an invasive species. The authors present well-documented data that Monk Parakeets are ecosystem engineers creating nesting sites for other species by virtue of their unusual nest structure and behavior.  The paper is well written and the analyses well structured.  The figures are useful (the photos added will help readers unfamiliar with Monk Parakeet nests).  Below are few questions and suggestions that will make the paper clearer.  

Thank you very much for these positive comments, we appreciate very much all the detailed suggestions provided. We have addressed all of them since we feel they are helping us to improve the clarity of our manuscript. Changes can be easily seen in the new version with tracked changes.

While the authors state that Monk Parakeets are one of the most widespread and successful invasive bird species, their success in terms of going outside of specific urban areas in most (all?) of the regions where they have become established pales in comparison to other bird species like European Starling, House Sparrow and Eurasian Collared Dove (at least from a North American perspective) so this language could be tempered a little. 

Compared with other invasive bird species, such as those mentioned by the reviewer, the monk parakeet shows a lower success of invasion. We therefore agree and have tempered that statement (line 58).

With respect to how were nests searched for and identified, do you worry about any spatial biases in sampling locations (specifically, data composed mostly of urban nests in invasive range and mostly rural nests in native range)? 

We understand the concern of the reviewer, but our study was part of other extensive field works performed in the Neotropics, across habitats with different degrees of urbanization (urban, rural, and natural). Our extensive censuses allow us to be confident that this is not a methodological bias: the monk parakeet is most often present in rural areas across its native range, while it has mainly established in urban areas in its invaded ranges. However, this asymmetry between the use of habitats does not affect our results, as we use the percentage of nests used by other species and their abundances. To clarify this point, we have added more information about our fieldwork procedures in Materials and Methods (lines 94-97).

Were nests in different regions of the world visited during comparable periods of time (other than the previously stated nests visited during the breeding season)? Do you think the species composition of tenants might be affected by seasonal changes in biodiversity and nest usage (due to migratory species)? 

Our systematic observations were only conducted during the breeding season of monk parakeets. Therefore, we could not assess potential seasonal changes in the biodiversity of tenants. Nonetheless, non-systematic observations in Spain indicate that some of the recorded tenant species can be present in monk parakeet nests throughout the year, such as house sparrows and rock pigeons. This apparent permanent occupation is probably because these species may lay multiple clutches per year. However, it is expected that the rest of tenant species with a single clutch do not use monk parakeets nests out of their breeding seasons, although shorty visits and prospecting behaviors can happen in monk parakeet nests during the non-breeding period, such as we have observed with rose-ringed parakeets and spotless starlings.

Other suggestions:

Line 32.  Replace "facilitate" with "present"

Done. As both reviewers provided different alternatives to change this word, we have chosen the option from the 1st reviewer.

Line 44. Keywords:  "nest inquilines" I looked up this term and can see why it is used, but it is never brought up anywhere else in the paper and I haven't seen it used with birds before.  I think it should be brought up and defined somewhere in the introduction.  

We agree and have incorporated its definition in the Introduction (lines 75-80), adding new references to support it:

  1. Riley, J.; Winch, J.M.; Stimson, A.F.; Pope, R.D. The association of Amphisbaena alba (Reptilia: Amphisbaenia) with the leaf-cutting ant Atta cephalotes in Trinidad. J. Nat. Hist. 1986, 20, 459-470.
  2. Hugo, H.; Cristaldo, P.F.; DeSouza, O. Nonaggressive behavior: A strategy employed by an obligate nest invader to avoid conflict with its host species. Ecol. Evol. 2020, 10, 8741-8754.
  3. Oschadleus, H.D. Birds adopting weaver nests for breeding in Africa. Ostrich 2018, 89, 131-138.
  4. Delhey, K. Nest webs beyond woodpeckers. Ecology 2018, 99, 985-988.
  5. van der Hoek, Y.; Gaona, G.V.; Ciach, M.; Martin, K. Global relationships between tree-cavity excavators and forest bird richness. R. Soc. Open Sci. 2020, 7, 192177.

Also: are many of these "cavity nesters" or species that like to place nests in cavities (e.g., Passer)?  There may not be a better term here.

Near 50% of the recorded tenant species are cavity nesters, and most of the recorded nests (> 80%) correspond to these species, so we have added the percentages in Results (lines 176-177). Regarding the term "cavity nester", we think this term is correct because we have avoided considerer as cavity nesters those species that marginally use cavities. In the case of birds of the genus Passer, all recorded species in this study but Passer hispaniolensis are cavity nesters.

Line 64: “Human utilities”, should be "human-made structures"  or "human construction"

Done.

Line 87.  Change "areas" to "ranges"

Done.

Line 89: As Line 64, should be human structure or man-made structure .

Done.

Line 98.  I think the word "conservatively" can be deleted here.

Done.

Line 106.  Say where the photo was taken.

Done.

Line 121.  Change ", obtaining thus" to "which resulted in..." 

Done.

Lines 124-126: Contrary to Lines 93-95, it seems to be implied that nests in Puerto Rico were also sampled across multiple years, in order to be included in the persistence analysis. If so, what years were sampled?  

In the case of Puerto Rico, we have sampled only one breeding season. We have specified the study areas where we recorded the persistence of nests across years (line 145).

Line 127: What is meant by “roof” as a substrate? Were nests built on top of ledges, on windowsills, under bridges, (and other man-made structures that are not pylons) also labeled as roof? I would suggest rewording to use a clearer term for “roof” or an in-text clarification of what this category encompasses. 

We have clarified categories "pylon" and "roof" in Materials and Methods (lines 101-103).

Lines 131-132: …between monk parakeets and their tenants, and monk parakeets and their predators.

Done. 

Line 166.  This could be made a little clearer (and maybe tested).  Were there likely to be more species using a nest structure if species had different needs (nest types, one.

We are sorry we do not see the way to make these descriptive results clearer, nor the need for adding further tests.

Line 266: Should the word cooperative be used instead of “facilitative”? 

We think the word "facilitative" is correct because the word "cooperative" would fit when both tenant and host species are benefited, while the facilitation corresponds not only to this interaction but also when only one of those species is benefited. Our study shows monk parakeets benefitting or not during interspecific interactions depending on the tenant species.

Line 315. Change to "parakeets"

Done.

Line 318.  Change this to "coincides with monk parakeet population growth and expansion.

Done.

Line 320.  Any idea why these species are declining?

In the case of stock doves (Columba oenas), factors such as the low availability of tree cavities after the reduction of mature trees as well as the hunting pressure, are the main causes of its decline. In addition to the low availability of tree cavities, Western jackdaws (Coloeus monedula) were intensively persecuted as "pests".

We have added a recent study focused on a stock dove population from UK that also supports the reduction of mature trees and, consequently, available cavities, as a limiting factor for its population growth.

New reference:

  1. Richardson, J.E.; Lees, A; Marsden, S. Landscape -scale habitat associations in an urban Stock Dove Columba oenas popula-tion. Res. Sq. (Preprint).

Line 324.  Make the context of this clear, are you meaning if management plans dictate population control measures?

We have clarified this point (lines 349-350).

Table 1.  Make the distinction between invaded and native areas clear in the heading.

Accordingly, we have changed the format of the table header.

Figure 2 is referenced in Line 191 to show that invasive populations had a higher presence, abundance, and richness of tenants than the native range. The graphs in the figure do not make that point obvious (seeing all the overlap among the confidence intervals).  This figure could be moved to the supplemental data.

We agree the figure was not clear. We have redone it for clarity (now only focusing on native and invasive areas), as we feel it is important to keep it in the main body of the text. The previous version of Figure 2 that shows all invasive populations (Puerto Rico, Continental Spain, Tenerife) has been moved to the supplemental data.